# Global burden and trends of age-related and other hearing loss: A 32-year analysis and future projections based on the GBD 2021

**Jiao Zhu, Min Yang, Cuiying Zhou, Houyong Kang, Deping Wang**◉*

Department of Otorhinolaryngology, The Second Affiliated Hospital of Chongqing Medical University, Chongqing, China

* wangdepingent@hospital.cqmu.edu.cn

## Abstract

### Background

To evaluate the global, regional, and national burdens of and trends in age-related and other hearing loss (ARoHL) from 1990–2021 based on the Global Burden of Disease 2021 database.

### Methods

This study examined trends and disparities in the prevalence and years lived with disability (YLDs) of patients with ARoHL across age, sex, and the sociodemographic index (SDI). The estimated annual percentage change (EAPC) was calculated to assess temporal trends. Decomposition analysis, cross-country inequality analysis, and frontier analysis were employed to reveal additional facets of the ARoHL burden, whereas Bayesian Age-Period-Cohort (BAPC) modeling projected future trends to 2040.

### Results

ARoHL remains a critical public health challenge. The global age-standardized prevalence rate (ASPR) increased significantly from 1.71 (95% UI: 1.63–1.80) ×10⁴ to 1.81 (95% UI: 1.73–1.89) ×10⁴ per 100,000 (EAPC = 0.163; 95% CI: 0.154–0.172), whereas the age-standardized YLD rate (ASYR) increased from 499.37 to 525.87 per 100,000 (EAPC = 0.171; 95% CI: 0.161–0.180). Decomposition analysis revealed that epidemiological changes contributed 37.28% to the increase in YLDs. Globally and across all five SDI regions without age distinction, the male ASPR and ASYR were consistently greater than the female ASPR and ASYR at all time points. The relationship between the SDI and ARoHL burden is complex. BAPC projections indicate stable ASPRs and ASYRs through 2040 despite increasing cases and YLDs.

**Data availability statement:** The minimal datasets underlying this study are publicly available from the Global Burden of Disease (GBD) Results tool hosted by the Institute for Health Metrics and Evaluation (IHME) (https://vizhub.healthdata.org/gbd-results). No additional approvals, credentials, or contact with IHME are required beyond the free registration. By following the identical parameter settings and analytical methods described in the Methods section, other researchers can fully replicate all findings reported in this study. The authors had no special access privileges. All data were obtained through the same public interface and under the same terms available to any registered user.

**Funding:** The author(s) received no specific funding for this work.

**Competing interests:** The authors have declared that no competing interests exist.

**Abbreviations:** ARoHL, Age-Related and other Hearing Loss; ASPR, Age-Standardized Prevalence Rate; ASR, Age-Standardized Rate; ASYR, Age-Standardized Ylds Rate; BAPC, Bayesian Age-Period-Cohort; CI, Confidence Interval; EAPC, Estimated Annual Percentage Change; GHDx, Global Health Data Exchange; ICD, International Classification of Diseases; SDI, Sociodemographic Index; SII, Slope Index of Inequality; UI, Uncertainty Intervals; YLDs, Years Lived with Disability.

## Conclusions

The global ASPR of ARoHL increased by 5.63% and that of ASYR increased by 5.31% from 1990–2021, with the number of cases and YLDs doubling. Targeted interventions and policies must address this growing public health challenge.

## Introduction

Hearing loss is a significant public health concern that affects people of all ages and places a considerable burden on health care systems and economies worldwide [1,2]. Age-related and other hearing loss (ARoHL), a major category of hearing loss, is particularly noteworthy and contributes significantly to the overall health burden. In the Global Burden of Disease (GBD) study, ARoHL is categorized as a Level 3 cause, which includes conditions not attributed to congenital hearing loss, chronic otitis media, or meningitis. The predominant contributor to ARoHL is presbycusis, characterized by a gradual age-related decline in hearing due to the degeneration of inner ear neurons [3].

ARoHL has a substantial effect on health and quality of life; however, it remains inadequately addressed in many regions, particularly in low- and middle-income countries [4]. Previous GBD analyses have shown that hearing loss is one of the leading causes of disability globally, with ARoHL significantly contributing to its increasing prevalence and impact [5,6]. However, these previous analyses have limitations, such as the use of outdated data and a focus on different populations and analytical perspectives. These limitations highlight the need for more detailed and up-to-date analyses to inform targeted interventions and policy-making.

To address these gaps, our study employs the most recent data from the GBD 2021 to provide a comprehensive analysis of the global, regional, and national burdens of ARoHL. We conducted a series of analyses, including disease burden estimation, decomposition analysis, cross-country inequality analysis, frontier analysis, and predictive analysis, to elucidate key trends and disparities in prevalence and years lived with disability (YLDs) and to identify the underlying drivers of this burden, as well as potential areas for improvement in managing ARoHL. These analyses provide valuable insights for policy-makers, health care providers, and stakeholders involved in hearing loss prevention and rehabilitation, ultimately supporting the development of evidence-based strategies to mitigate the growing burden of ARoHL.

## Materials and methods

### Data acquisition

This study conducted a systematic analysis using data from the GBD 2021 database. As an internationally recognized authoritative platform for epidemiological research, the GBD 2021 database integrates multidimensional epidemiological data from 204 countries and regions worldwide, covering 371 diseases and health issues, as well as 88 risk factors, spanning the years 1990–2021 [7]. Focusing on ARoHL, this study obtained refined data resources through the Global Health Data Exchange (GHDx;

accessible at https://vizhub.healthdata.org/gbd-results/), with a particular emphasis on analyzing the distribution charac-teristics of core indicators such as prevalence and YLDs across age, gender, and geographical dimensions. To further explore how socioeconomic factors affect disease burden, this study employed the sociodemographic index (SDI) evalu-ation framework (data source: https://ghdx.healthdata.org/gbd-2021). This index synthesizes three key parameters—per capita income, educational attainment, and the fertility rate—to categorize study regions into five developmental tiers (from low to high), providing a quantitative analytical tool for revealing the heterogeneous distribution patterns of ARoHL in resource-scarce regions and aging societies. Notably, the SDI framework not only effectively reflects regional develop-mental disparities, but it also facilitates the assessment of the efficiency of health care resource allocation, the effective-ness of public health policy implementation, and the prioritization of disease prevention and control. This, in turn, lays a scientific foundation for the formulation of targeted intervention strategies.

## Data sources and disease model

The GBD 2021 study systematically integrates epidemiological surveys, hospital records, insurance claims data, and published literature from 204 countries and regions worldwide. The GBD research network employs standardized meth-ods for data cleaning, correction, and modeling to ensure the comparability and reliability of the data. The definition of ARoHL is based on the International Classification of Diseases (ICD) coding system, which primarily includes H91.1 (Presbycusis) and H90-H91 (other hearing loss-related codes) from the ICD-10, as well as corresponding codes from the ICD-9 to support historical data analysis. For modeling, the Bayesian regression model (DisMod-MR 2.1) was utilized to model global hearing loss data by estimating core indicators such as prevalence and YLDs. The model incorporates covariates such as age, sex, region, and the SDI and assesses data quality and the robustness of model results through uncertainty analysis. Additionally, this study employed a meta-regression analysis to interpolate data for regions with missing information, ensuring the completeness and representativeness of the global disease burden assessment. These findings provide a scientific basis for revealing the epidemiological characteristics of hearing loss and its trends over time. Additional modeling details are provided in the GBD 2021 methods appendices (https://www.healthdata.org/gbd/methods-appendices-2021).

## Estimation of disease burden

This study employs a multidimensional analytical approach to systematically assess the global disease burden of ARoHL. To ensure comparability across diverse populations and temporal scales, all epidemiological rates are expressed per 100,000 population using two core standardized metrics: the age-standardized prevalence rate (ASPR) and the age-standardized YLD rate (ASYR). Both the ASPR and the ASYR were calculated using the direct standardization method, which adjusts for differences in age structure by applying age-specific rates from each country to a standard population [8]. Absolute counts are labeled "number of cases/YLDs". On the one hand, the ASPR reflects the distribution character-istics of the disease in populations, whereas on the other hand, the ASYR quantifies the actual impact of the disease on population health. To gain a deeper understanding of the distribution characteristics of disease burden and its influencing factors, this study further explores the differentiated impacts of demographic and socioeconomic factors, such as age, sex, and the SDI, on the global and regional distributions of disease burden.

In terms of trend analysis, this study adopts a longitudinal approach, utilizing long-term data from 1990–2021 and calculating the estimated annual percentage change (EAPC) to assess the temporal trends in the ASPR and ASYR specifically. The calculation of the EAPC is based on a log-linear regression model applied to these age-standardized rates, with the core formula being ln(ASR) = a + bx + e, in which ln(ASR) represents the natural logarithm of the ASPR or ASYR, x denotes the year variable, a is the intercept, b is the regression coefficient, and e is the error term. On the basis of the regression coefficient b, the EAPC is calculated as EAPC = (exp(b) – 1) × 100. To evaluate the statistical signifi-cance of the trend changes, the 95% confidence interval (CI) was calculated. When the upper limit of the CI is less than

0, it indicates a significant decreasing trend in the ASPR or ASYR; when the lower limit is greater than 0, it suggests a significant increasing trend. This quantitative analytical approach not only accurately captures the long-term changes in age-standardized burden measures but also provides crucial evidence for the development of public health policies [9].

## Decomposition analysis

This study employs a decomposition analysis approach to thoroughly examine the driving factors behind changes in the disease burden of ARoHL. It systematically assesses how demographic shifts, population growth, and the evolution of epidemiological characteristics affected disease burden between 1990 and 2021. On the basis of the theoretical framework proposed by Das Gupta, we constructed a multifactorial decomposition model capable of effectively isolating and quantifying the independent effects of key factors such as population aging, population growth, and epidemiological changes while controlling for the influence of other variables [10]. The results of the present study indicate that population aging has significantly intensified the disease burden of ARoHL, while the natural increase in population size has also contributed to the expansion of the affected population base. Notably, changes in epidemiological characteristics, including the evolution of pathogen resistance, improved health care access, and the widespread adoption of preventive measures, have had complex effects on the disease burden. Through quantitative analysis, this study not only identifies the specific contributions of each factor to the burden of ARoHL, but it also reveals the interaction mechanisms between these factors. These findings provide scientific evidence for a comprehensive understanding of the epidemiological evolution of ARoHL and lay a critical data foundation for public health agencies to develop targeted preventive interventions and resource allocation strategies. The application of this analytical approach offers a methodological paradigm that can be adapted for the study of the disease burden of other infectious diseases.

## Cross-country inequality analysis

Health inequality monitoring is a critical cornerstone for building an equity-oriented public health system. Its core lies in identifying and quantifying health disparities among different population groups, thus providing scientific evidence for the development of targeted intervention strategies. This study adopts a multidimensional inequality assessment framework and systematically explores the association between the global disease burden of ARoHL and the SDI through two core indicators: the slope index of inequality (SII) and the concentration index [11]. The SII is calculated using cross-sectional data at the national level, where a regression model between the ARoHL ASPR or ASYR and SDI is constructed using the weighted least squares method. This effectively captures the absolute health disparities between countries at different stages of development. Moreover, the concentration index is calculated using Lorenz curve analysis, where the population is stratified into quintiles on the basis of socioeconomic status, enabling an in-depth examination of the distribution of the disease burden of ARoHL across socioeconomic gradients. This dual-indicator complementary analysis strategy not only provides a comprehensive reflection of the macro patterns of health inequality but also reveals its microdistribution trends, offering essential quantitative support for formulating targeted health equity promotion policies. The results of this study have significant practical implications for optimizing resource allocation and reducing health disparities. Additionally, this study provides a methodological reference for health inequality research in other disease areas.

## Frontier analysis

This study adopts frontier analysis methods to construct an evaluation model based on the ASPR and ASYR that aimed to deeply explore the association mechanisms between the disease burden of ARoHL and the SDI. Compared with traditional linear regression models, frontier analysis offers significant advantages: it not only identifies the multidimensional driving factors influencing disease burden, but it also effectively captures the complex nonlinear relationship between the SDI and disease burden. Methodologically, this study innovatively combines local weighted regression (LOESS) with local polynomial regression. By setting a smoothing span parameter of 0.3, a nonlinear boundary curve between the SDI and ASPR/ASYR

was constructed. This boundary curve has important theoretical significance, as it establishes the theoretically achievable minimum ASPR/ASYR for each country or region at its specific development level, providing a scientific benchmark for evaluating the disease prevention and control performance of various countries. Additionally, by quantifying the gap between the actual observed disease burden and the theoretical minimum, this method allows for the precise identification of areas and directions that require focused improvement [12]. To ensure the robustness and reliability of the results, we conducted 1,000 bootstrap resampling iterations and calculated the average ASPR/ASYR corresponding to each SDI value, effectively controlling for data fluctuations. During the evaluation phase, we constructed a quantifiable evaluation index system by measuring the absolute distance between the actual ASPR/ASYR values of countries in 2021 and the boundary curve. This index not only reflects the current effectiveness of disease prevention and control in various countries, but it also effectively assesses their potential for reducing the disease burden of ARoHL in the future. This multidimensional evaluation method provides crucial scientific evidence for the formulation of targeted public health intervention strategies.

### Predictive analysis

This study used a Bayesian Age–Period–Cohort (BAPC) analysis model to predict the evolving trends of the disease burden of ARoHL. The predictive model was implemented on the R language platform, utilizing the integrated nested Laplace approximation (INLA) algorithm and the BAPC package, which efficiently handle complex hierarchical data structures. This model simultaneously incorporates three key dimensions—age effects, period effects, and cohort effects—not only capturing disease risk characteristics across different age groups but also reflecting the impact of social environmental changes and medical advancements on the disease spectrum [13]. Compared with traditional predictive methods, the BAPC model offers greater explanatory power and predictive accuracy, particularly in long-term trend analysis and inter-generational comparisons. This method provides a reliable analytical tool for understanding the epidemiological characteristics of diseases such as ARoHL in detail. On the basis of this modeling framework, we have predicted the changes in the age-standardized rates and absolute counts of the prevalence and YLDs of ARoHL between 2022 and 2035, which will serve as important evidence for optimizing health care resource allocation and formulating disease prevention strategies.

### Statistical analysis

This study uses standardized epidemiological indicators to quantify the disease burden, with the ASPR and ASYR calculated per 100,000 population. Additionally, the 95% uncertainty interval (UI) is reported to reflect the precision of the predicted results. For the key indicator, the EAPC, we provide the 95% CI. The entire data analysis and visualization process was conducted in R Studio (Windows platform, version 4.3.3). For statistical inference, we used a two-sided test, with a p value of less than 0.05 considered the threshold for statistical significance.

### Ethics statement

This study utilized publicly available, de-identified data from the GBD 2021 study. The University of Washington Institutional Review Board approved a waiver of informed consent for the use of such de-identified data in the GBD study. As the data used in this study are publicly available and do not contain any personally identifiable information, neither written nor verbal informed consent was required. All procedures performed in this study were in accordance with the relevant ethical standards and regulations, including those of the Declaration of Helsinki.

## Results

### Global and regional burden of age-related and other hearing loss

Globally, both the ASPR and ASYR of ARoHL increased significantly from 1990–2021. The global ASPR of ARoHL increased by 5.63% and the ASYR increased by 5.31% from 1990–2021, with the number of cases and YLDs doubling.

Specifically, the ASPR increased from 1.71 (95% UI: 1.63–1.80) × $10^4$ per 100,000 in 1990 to 1.81 (95% UI: 1.73–1.89) × $10^4$ per 100,000 in 2021, with an EAPC for the ASPR of 0.163 (95% CI: 0.154–0.172) (S1 Table in S1 File). Concurrently, the number of cases increased by 109%, from 7.41 (95% UI: 7.06–7.80) ×$10^8$ to 1.55 (95% UI: 1.48–1.62) ×$10^9$. Similarly, the ASYR increased from 499.37 (95% UI: 346.66–694.02) per 100,000 in 1990 to 525.87 (95% UI: 364.24–731.97) per 100,000 in 2021, with an EAPC for the ASYR of 0.171 (95% CI: 0.161–0.180) (S2 Table in S1 File). Moreover, the number of YLDs increased from 2.13 (95% UI: 1.46–2.96) ×$10^7$ to 4.44 (95% UI: 3.07–6.20) ×$10^7$.

Among the five SDI regions, the low–middle SDI region was the only region with an EAPC for the ASPR whose 95% CI spanned zero, suggesting an uncertain or nonsignificant trend. In contrast, the remaining four SDI regions all presented positive EAPCs for the ASPR, indicating a consistent upward trend in prevalence. Notably, the high–middle SDI region had the highest EAPC for the ASPR, at 0.279 (95% CI: 0.271–0.287). Additionally, the middle SDI region had the highest ASPR, with values of 1.91 (95% UI: 1.82–2.02) ×$10^4$ per 100,000 in 1990 and 1.97 (95% UI: 1.89–2.07) ×$10^4$ per 100,000 in 2021 (S1 Table in S1 File). Unlike the ASPR trends, the EAPCs for the ASYR were negative in the low-middle and low SDI regions, whereas the other three SDI regions presented positive trends (S2 Table in S1 File).

With respect to the 21 GBD superregions, only four—Central Sub-Saharan Africa, Oceania, Western Sub-Saharan Africa, and high-income North America—showed a declining trend in the ASPR, as indicated by their negative EAPCs. For Tropical Latin America, the EAPC for the ASPR had a 95% CI that included zero, reflecting an inconclusive trend. The remaining 16 superregions presented positive EAPCs for the ASPR, suggesting a continued increase in the ASPR. Among these, East Asia had the highest ASPR and number of cases. The ASPR in East Asia also increased from 2.06 (95% UI: 1.95–2.19) ×$10^4$ per 100,000 in 1990 to 2.19 (95% UI: 2.08–2.31) ×$10^4$ per 100,000 in 2021, with an EAPC for the ASPR of 0.203 (95% CI: 0.185–0.222). Specifically, East Asia alone accounts for nearly 30% of cases worldwide, reaching 4.59 (95% UI: 4.34–4.86) ×$10^8$ in 2021 (S1 Table in S1 File). Moreover, according to the EAPC for the ASYR, the ASYR increased in 8 superregions, remained statistically nonsignificant in 3 superregions, and decreased in 10 superregions (S2 Table in S1 File).

The temporal trends of the ASPR and ASYR are visually presented in S1 Fig in S1 File. The high SDI region presented the lowest ASPR and ASYR among the five SDI regions, with relatively stable trends over time. Although the overall trend of the ASPR indicated an upward trajectory from 1990–2021, it was accompanied by notable fluctuations over the past 32 years, particularly in low SDI regions and especially among females. Similarly, significant fluctuations were observed in female ASYRs within low SDI regions.

### Age–sex distributions of age-related and other hearing loss

Globally and across all five SDI regions without age distinction, the male ASPR and ASYR were consistently greater than the female ASPR at all time points, as shown in S1 Fig in S1 File. When analyzed by age group, the ASPR increased with advancing age globally, whereas the ASYR showed a similar upward trend but declined slightly among individuals aged 90 years and older (Fig 1). Males had a higher ASPR than females did in most age groups. In terms of absolute counts, the number of cases in males peaked between 55 and 59 years of age, whereas it peaked between 65 and 69 years of age in females. Both sexes presented the highest YLDs at 65–69 years of age. Males had a greater number of cases and YLDs than females did among individuals under 70 years of age, but females had a greater number of cases and YLDs at 70 years and older.

### Correlation between the burden of age-related and other hearing loss and the SDI

Over the past 32 years, across 21 GBD superregions, the ASPR showed a significant negative correlation with the SDI (R = −0.4109, P < 0.001); this negative correlation was stronger for the ASYR (R = −0.6774, P < 0.001). Notably, the relationship between the ASPR and SDI was nonlinear, with an initial increase followed by a decrease as the SDI increased (Fig 2). Similarly, across 204 countries and territories, both the ASPR and ASYR exhibited significant negative correlations with

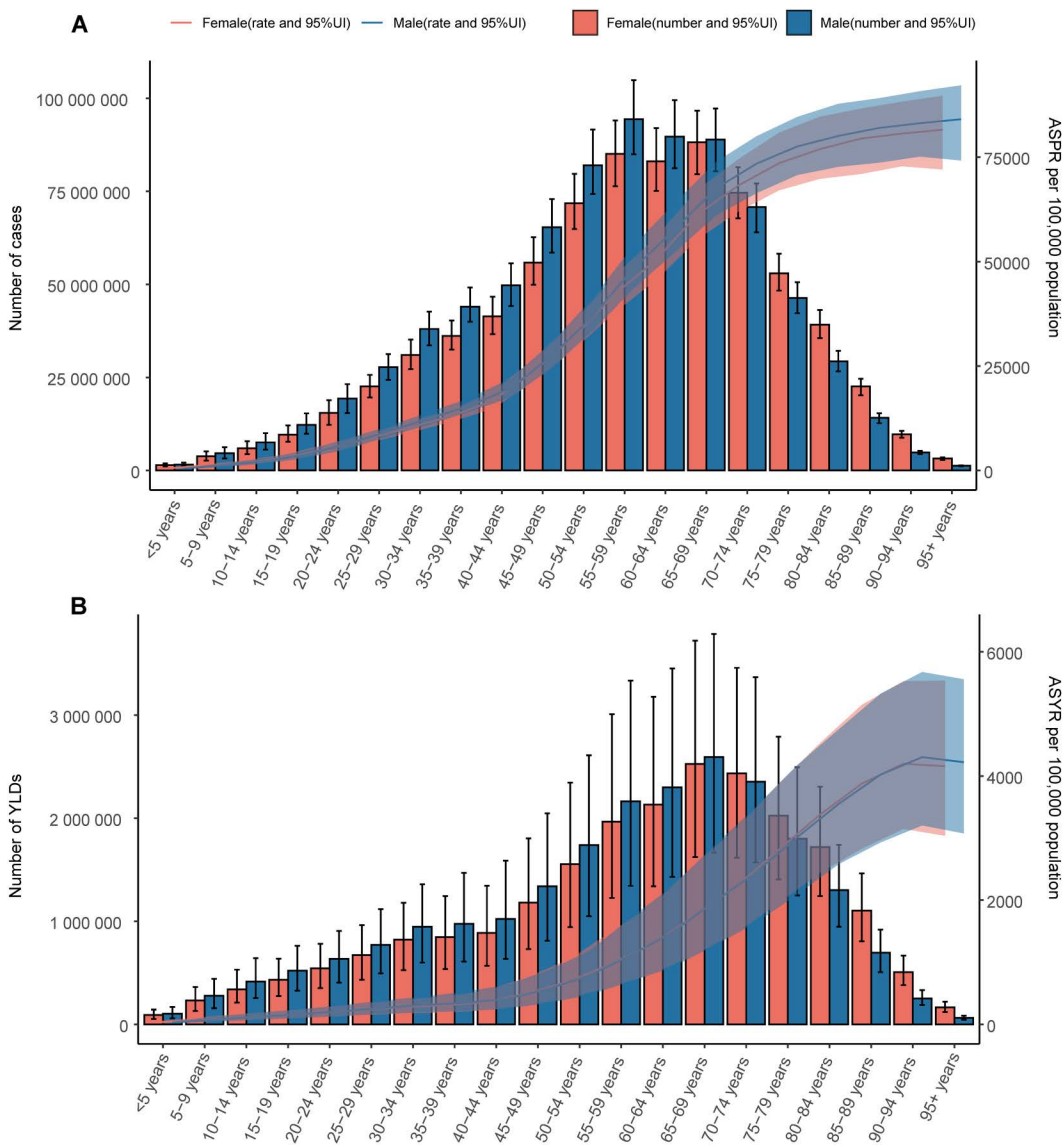

**Fig 1. Global age–sex distributions of the prevalence and YLDs of ARoHL in 2021.** This figure illustrates the global age–sex distribution of the prevalence of ARoHL and the YLDs in 2021, displaying the ASPR/ASYR per 100,000 and the number of cases/YLDs for different age groups and sexes. The lines indicate the ASPR/ASYR per 100,000, with shaded areas indicating the 95% uncertainty interval. The bars represent the number of cases/YLDs. Red represents females, and blue represents males. (A) Number of cases and the ASPR; (B) YLDs and the ASYR.

the SDI (ASPR vs. SDI: R=−0.2775, P<0.001; ASYR vs. SDI: R=−0.5942, P<0.001) (S2 Fig in S1 File). However, these correlations were not entirely linear, indicating complex relationships between these variables.

Further analysis of the 204 countries and territories revealed various trends. The ASPR showed a weak positive correlation with its EAPC from 1990–2021 (R=0.25, P=0.00025), characterized by a wavelike trend line that indicates fluctuating changes in the rate of change in the ASPR over time (S3 Fig in S1 File). In contrast, the correlation between the SDI and the EAPC for the ASPR was negligible (R=0.0064, P=0.93), with a nearly flat trend line. Moreover, the SDI exhibited a weak positive correlation with the EAPC for the ASYR (R=0.14, P=0.049). However, no significant correlation was found between the ASYR and its EAPC (R=−0.071, P=0.31), with a valley-shaped trend line.

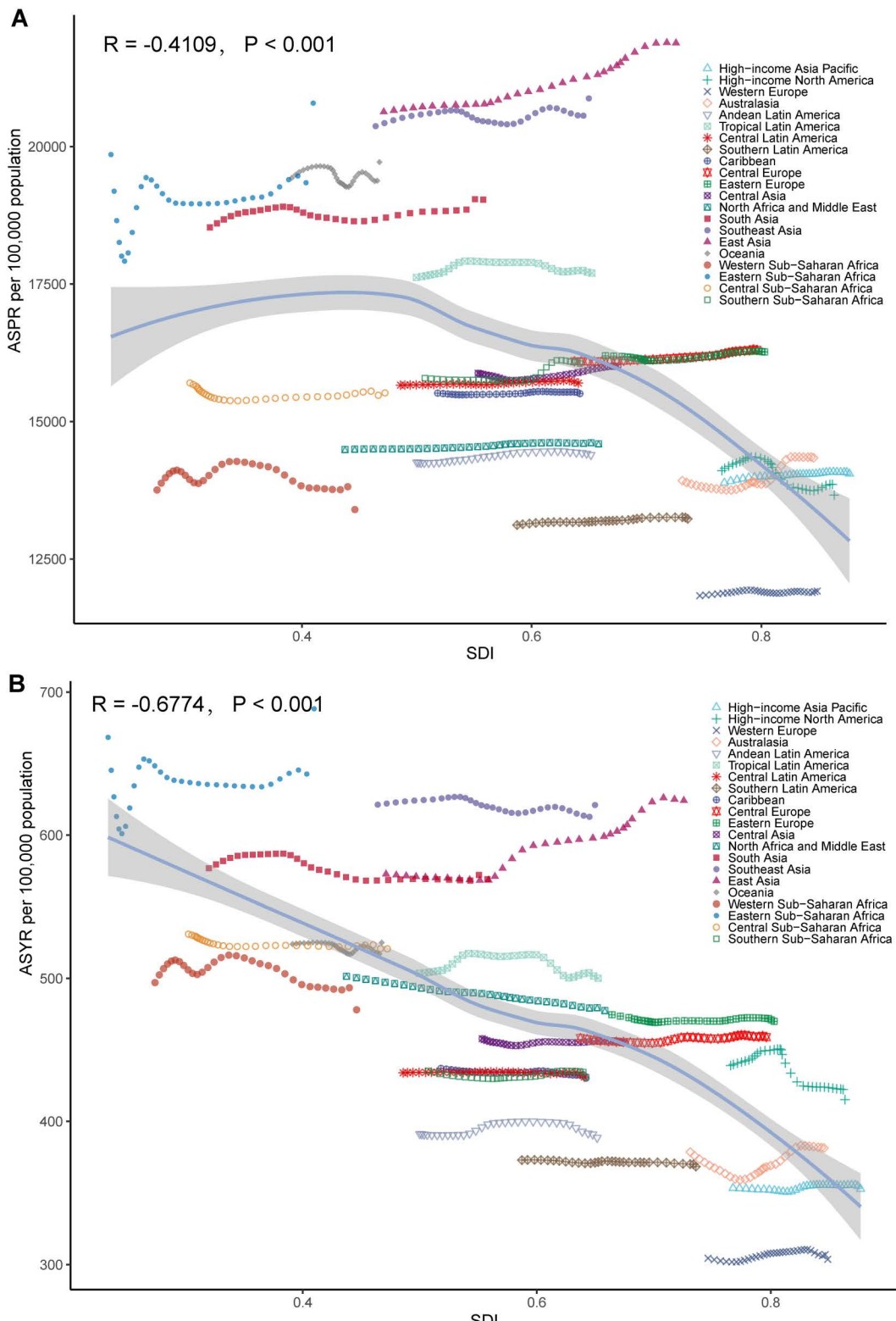

**Fig 2. Correlation between the SDI and the ASPR/ASYR of ARoHL across 21 GBD superregions.** This figure shows the correlation between the SDI and the **ASPR/ASYR** of ARoHL across 21 GBD superregions. Different colored shapes represent different GBD superregions. The blue curve shows the relationship between the SDI and the **ASPR/ASYR**, with the shaded area indicating the 95% confidence interval. (A) ASPR vs. SDI; (B) ASYR vs. SDI.

## Geographical distribution of age-related and other hearing loss burdens

The distributions of the ASPR and ASYR in 2021, as well as their EAPCs from 1990–2021, across 204 countries and territories are shown in S3 and S4 Tables in S1 File. In 2021, the highest ASPR and ASYR were recorded in the Republic of Madagascar, with an ASPR of 22,034.32 (95% UI: 20,795.57–23,519.21) per 100,000 and an ASYR of 782.67 (95% UI: 555.57–1,075.28) per 100,000. Conversely, the lowest ASPR and ASYR were observed in the Kingdom of Sweden, with an ASPR of 10,107.35 (95% UI: 9,655.64–10,601.05) per 100,000 and an ASYR of 243.21 (95% UI: 162.96–347.40) per 100,000. For the period from 1990–2021, the five countries with the highest EAPCs for the ASPR were the Federal Democratic Republic of Ethiopia (0.226; 95% CI: 0.131–0.321), the Republic of Uganda (0.221; 95% CI: 0.081–0.361), the People's Republic of China (0.208; 95% CI: 0.189–0.227), the Republic of Equatorial Guinea (0.196; 95% CI: 0.155–0.238), and the Republic of Mozambique (0.193; 95% CI: 0.083–0.303). The highest EAPCs for the ASYR were observed in the People's Republic of China (0.345; 95% CI: 0.306–0.385), the Federal Democratic Republic of Ethiopia (0.159; 95% CI: 0.066–0.253), Australia (0.155; 95% CI: 0.071–0.239), the United Republic of Tanzania (0.138; 95% CI: 0.027–0.249), and an unspecified country (0.125; 95% CI: −0.023 to 0.274).

## Decomposition analysis

The decomposition analysis revealed that the increase in the number of cases worldwide is attributed primarily to population growth and aging (Fig 3). Specifically, population growth accounted for 53.10% of the increase, whereas aging contributed to 39.15% of the increase. Epidemiological changes, such as increased exposure to risk factors, accounted for 7.75% of the increase (S5 Table in S1 File).

The contribution of epidemiological changes varied significantly among the five SDI regions. In the high–middle SDI regions, epidemiological changes accounted for 12.95% of the increase in the number of cases, which was the highest proportion among all the SDI regions. Conversely, in the low SDI regions, aging had the most substantial impact, contributing 52.39% to the increase in the number of cases. In the high SDI regions, population growth was the dominant factor, accounting for 72.37% of the increase in the number of cases. Despite the smaller role epidemiological changes played in the increasing numbe  of cases, there was still a positive correlation.

Population growth was the main driver for the increase in YLDs, contributing to 62.72% of the global increase. Epidemiological changes accounted for 37.28% of the increase, whereas aging had a negligible effect, contributing to only −0.01% of the increase. These findings indicate that while population growth is a significant driver of the overall burden, the effect of aging on YLDs is minimal at the global level.

## Health inequality analysis

A regression analysis revealed that the slope index of inequality of the ASPR decreased from −2505 in 1990 to −3134 in 2021 (Fig 4). The data points were widely dispersed and did not cluster closely around the regression line, with data from China and India having the most significant impact on the slope index of inequality. The concentration index of the ASPR was −0.04 (95% CI: −0.05 to −0.03) in 1990 and −0.01 (95% CI: −0.03 to 0.00) in 2021, highlighting a higher ASPR in countries with lower SDI values. Although the degree of inequality has diminished over time, some disparities persist.

For the ASYR, the relative rank by the SDI changed from −185 in 1990 to −174 in 2021, with no crossover between the two lines. These findings demonstrate that although the relative burden of the ASYR remains stable across countries with different SDI values, the overall burden has decreased. The concentration index for the ASYR was −0.07 (95% CI: −0.08 to −0.06) in 1990 and −0.04 (95% CI: −0.06 to −0.03) in 2021, highlighting a greater burden of the ASYR in lower SDI countries. Similar to the ASPR, the degree of inequality of the ASYR has decreased over time but still exists.

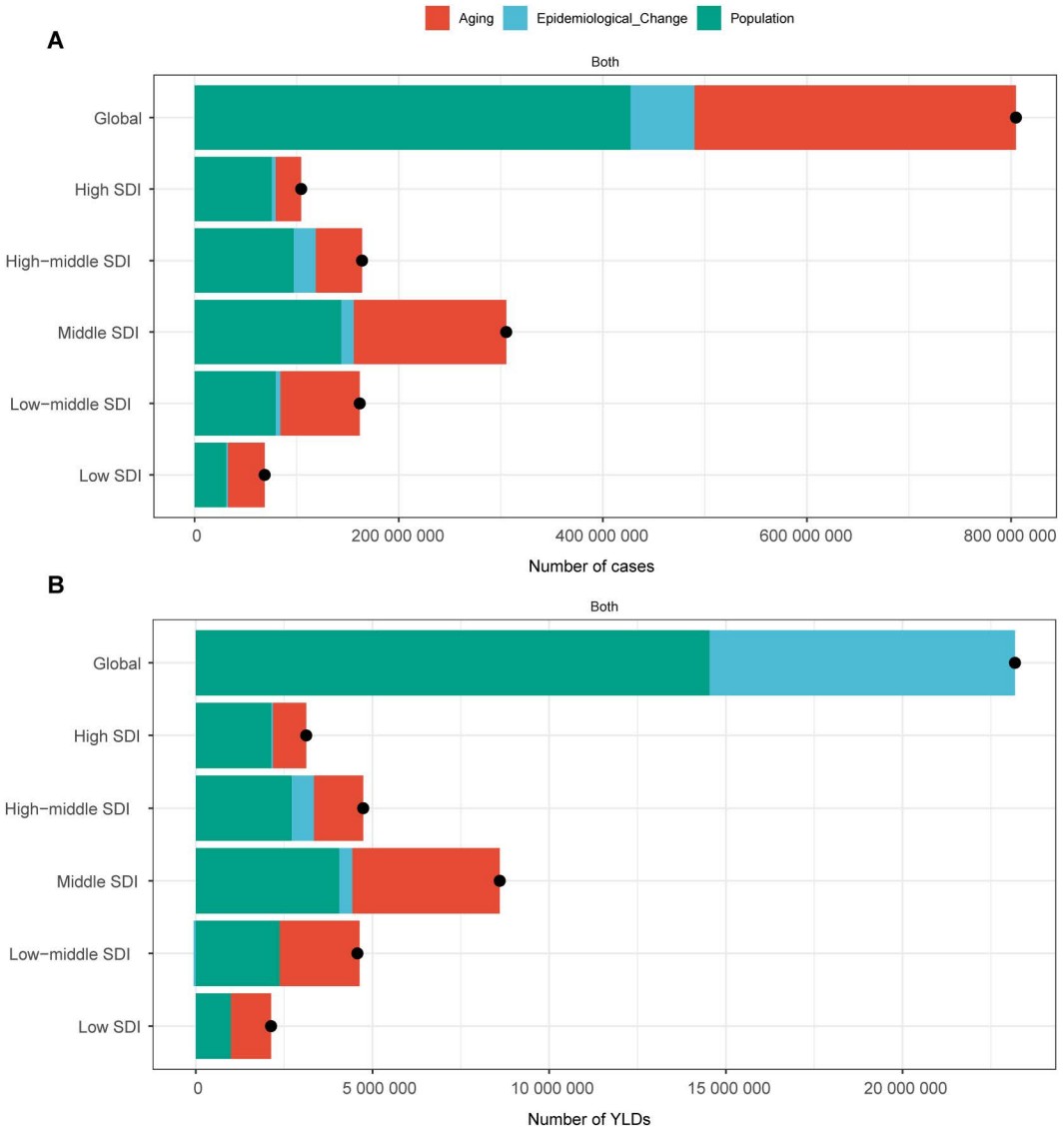

**Fig 3. Decomposition analysis of factors influencing the number of cases/YLDs of ARoHL.** This figure presents the results of the decomposition analysis, which assessed the effects of population growth, epidemiological changes, and aging on the number of cases/YLDs of ARoHL over the past 32 years. It highlights the contributions of each factor to the overall changes in the number of cases/YLDs. The black dots represent the cumulative changes resulting from the combined effects of all three factors. For each individual factor, a positive value reflects an increase in the relevant measure due to that factor, whereas a negative value indicates a decrease in the metric associated with that factor. (A) Number of cases; (B) Number of YLDs.

## Frontier analysis

In the context of the ARoHL burden, a frontier analysis was conducted to identify countries with the greatest potential for improvement (S4 Fig in S1 File). With respect to the ASYR, China, Maldives, Seychelles, Malaysia, Mauritius, the Philippines, Guam, Sri Lanka, Thailand, Madagascar, Vietnam, Indonesia, Malawi, the Northern Mariana Islands and the Cook Islands emerged as having the highest potential for reducing their ARoHL burden. With respect to the ASYR, countries with significant potential for improvement included Kenya, Madagascar, the Philippines, China, Maldives, Thailand,

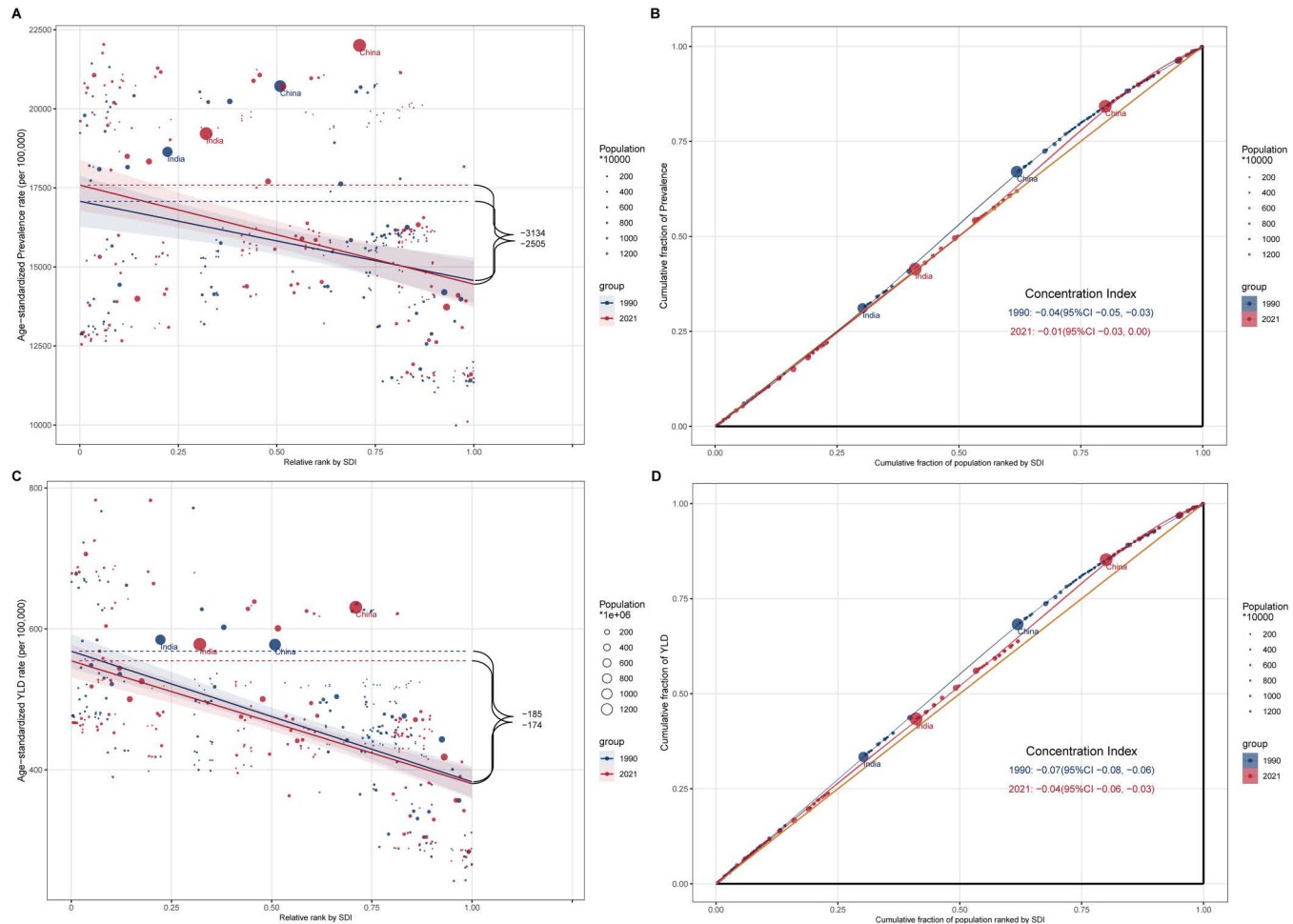

**Fig 4. Health inequality analysis.** This figure presents the health inequality analysis conducted using the slope index of inequality (SII) and the concentration indices of the ASPR and ASYR for ARoHL from 1990–2021. This highlights the disparities in the ARoHL burden among different socioeconomic groups, with a specific emphasis on populous countries such as China and India. Blue represents 1990, and red represents 2021. Each dot represents a different country or territory, with the size of the dot indicating the population of that country or territory. (A) Health inequality regression curves for the ASPR of ARoHL. (B) Concentration curves for the ASPR of ARoHL. (C) Health inequality regression curves for the ASYR of ARoHL. (D) Concentration curves for the ASYR of ARoHL.

Malaysia, Sri Lanka, Seychelles, Mauritius, Vietnam, Malawi, Indonesia, Ethiopia and Myanmar. These insights underscore the necessity for targeted public health interventions in these regions to effectively control ARoHL and enhance health outcomes.

## Bayesian age-period-cohort (BAPC) analysis

The BAPC model was employed to forecast the global burden of ARoHL through 2040 (Fig 5). Overall, projections show that the ASPR and ASYR are expected to remain stable globally. In contrast, the number of cases and YLDs are projected to increase globally. This trend is consistent across both sexes. Specifically, the number of cases is projected to increase to 2.24 (95% UI: 1.65–2.83) $\times 10^9$ globally. YLDs is expected to increase to 6.47 (95% UI: 4.64–8.31) $\times 10^7$.

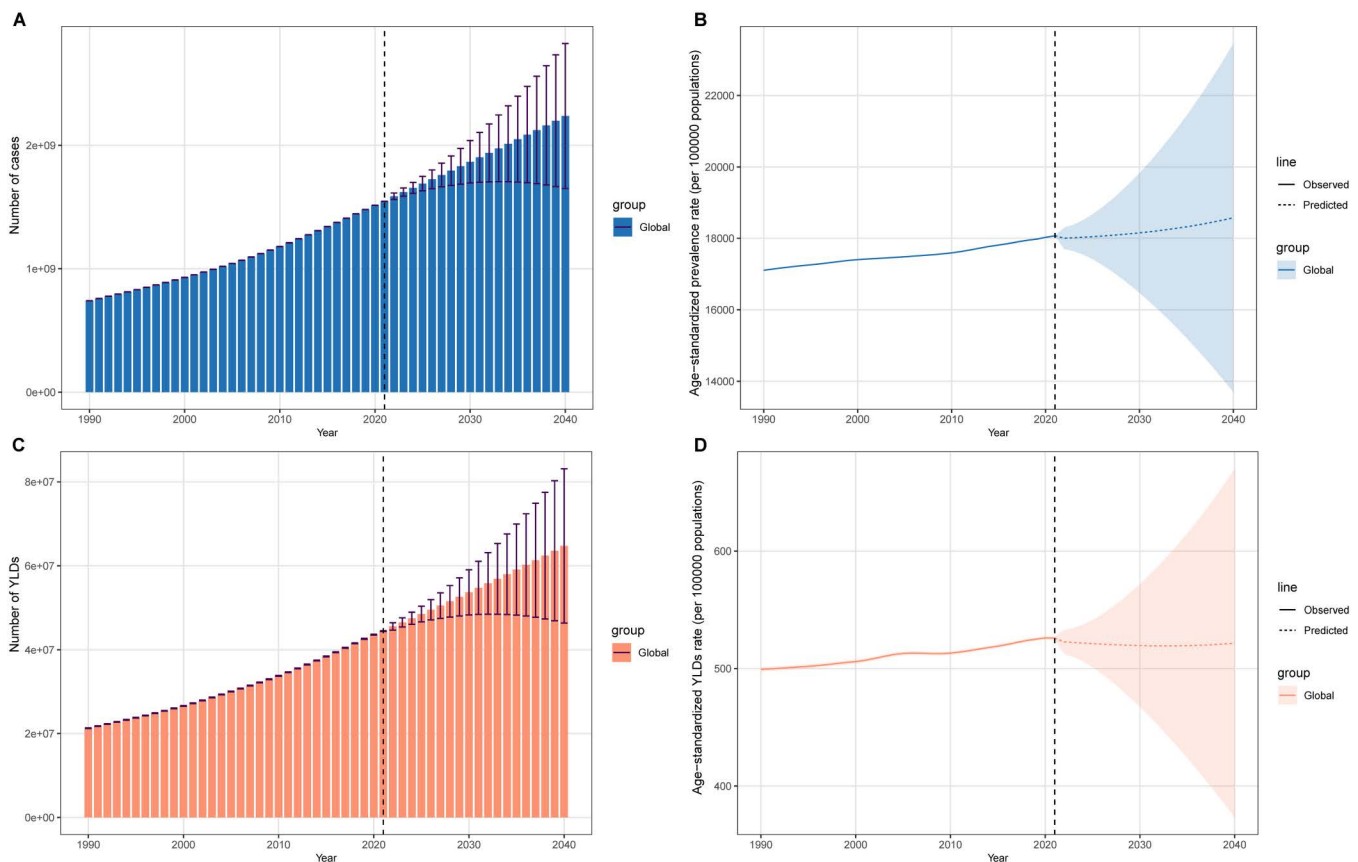

**Fig 5. Bayesian age–period–cohort (BAPC) analysis projections for the ARoHL burden, 1990–2040.** This figure presents BAPC model projections for global ARoHL burden indicators. The analysis includes projections for future trends, highlighting the expected changes in the ARoHL burden over time. This figure provides insights into the potential future effect of ARoHL on global health. (A) Number of cases; (B) ASPR; (C)Number of YLDs; (D) ASYR.

## Discussion

ARoHL remains a critical public health challenge. Over the past three decades, the global ASPR of ARoHL has increased by 5.63% (EAPC 0.163), that of ASYR has increased by 5.31% (EAPC 0.171), and the number of cases and YLDs have doubled. This significant increase in the number of cases is largely due to two factors that are difficult to change in the short term: population growth and aging. Thus, to reduce the number of ARoHL cases, a more feasible approach would be to improve treatment technologies and increase their accessibility, including expanding their availability and reducing costs. Since 2020, the significant decline in cochlear implant costs has made widespread implementation more feasible, yet its realization now largely depends on political will [14]. As emphasized by the WHO, integrating hearing health services into universal health coverage is essential to achieving "hearing health for all" [2]. Of course, as the current mainstream treatment options, hearing aids and cochlear implants are not sufficient to meet all needs. Recent advancements in gene therapy for deafness have offered new hope for further treatment [15]. The development of new treatment methods and the integration of these methods with local policies are urgently needed to reduce the burden of ARoHL.

Another factor contributing to the increase in the number of cases is epidemiological changes, which account for 7.75% of the increase globally. Although this is not the primary driver, its impact is particularly notable in high–middle SDI regions, where it contributes to 12.95% of the increase. In contrast, epidemiological changes have a more pronounced

effect on the increase in YLDs globally, contributing to 37.3% of their increase and thereby emphasizing the importance of preventable risks. Environmental factors such as occupational noise and ototoxic chemical exposure play a significant role in rapidly industrializing economies, as highlighted by recent global occupational health studies [16]. These studies reveal that a considerable number of manufacturing workers worldwide are subjected to noise levels above 85 dB, which is a key preventable risk factor for hearing loss [17,18]. Given these findings, the need for tailored interventions is underscored, particularly in regions undergoing rapid industrialization without adequate health safeguards [19]. Addressing these factors through robust policy measures is crucial to effectively reduce the burden of ARoHL. Additionally, emerging evidence implicates hereditary susceptibility (e.g., GJB2 mutations prevalent in East Asia) and iatrogenic risks from otologic surgeries or head/neck radiation in accelerating ARoHL pathogenesis, necessitating genetic screening programs and clinical surveillance protocols [20–22].

Given the multifactorial nature of ARoHL, understanding the role of the SDI in shaping the burden of disease is essential for formulating effective policies. However, the relationship between the SDI and the ARoHL burden is intricate and multifaceted. Generally, higher SDI levels, which are indicative of superior economic conditions, more effective policies, and advanced health care systems, are associated with lower disease burdens. This is supported by cross-country inequality analyses and confirmed by negative correlations between the SDI and the ASPR/ASYR. However, this correlation is not strictly linear. The nonlinearity may be attributed to a transitional "epidemiological trap" in middle SDI regions [23]. In these regions, industrialization has led to increased noise exposure, which outstrips current preventive efforts, mirroring patterns observed in cardiovascular epidemiology [24]. This clearly underscores the importance of appropriate policies in reducing disease burden. Additionally, the relationship between the SDI and EAPCs for the ASPR/ASYR highlights that socioeconomic progress alone is insufficient to curb ARoHL without targeted policies. For example, China's high EAPC of the ASPR (0.208) reflects the challenges posed by rapid industrialization and the need for further improvements in noise control. In response, China has launched the National Hearing Health Action Plan to address these issues comprehensively [25]. This plan aims to improve hearing health through enhanced prevention, early detection, and rehabilitation services, reflecting the country's commitment to addressing the growing burden of hearing loss.

Moreover, sex has a significant effect on the burden of ARoHL. Our analysis reveals a gender-related paradox in ARoHL. Without age stratification, males exhibited consistently higher ASPRs and ASYRs than females did globally and in all five SDI regions. For absolute counts, males under 70 years of age had a greater number of cases and YLDs, whereas females had higher values at 70 years of age and older. This pattern is consistent with evidence indicating that males are exposed to significantly higher levels of occupational noise [18,26]. Additionally, the protective effects of estrogen in premenopausal females may delay age-related hearing decline [27–29]. The marked increase in the female ARoHL burden after the age of 70, which coincides with global gains in life expectancy, underscores the need for sex-specific rehabilitation strategies. Understanding these sex differences is crucial for developing targeted interventions. This is particularly important in low SDI regions, where the female ASPR fluctuates widely.

In addition to the current significant burden, BAPC projections indicate that the global burden of ARoHL will continue to pose ongoing challenges. Research has shown that the economic burden of hearing loss is considerable, amounting to nearly $980 billion annually [1]. Beyond these numbers lies a deeper reality: the personal experiences of individuals, the struggles of families, and the broader societal challenges that these statistics represent. The urgency of taking effective measures to address the current situation and reduce future burdens while enhancing overall human well-being cannot be overstated. Ensuring equitable access to hearing health services is not only an immediate public health priority but also an economic necessity [30]. In regions where the burden of ARoHL is expected to increase, targeted interventions and resource allocation will be crucial in mitigating the impact of ARoHL [31]. Recommended interventions include occupational noise control in rapid-burden regions, gender-specific rehabilitation for older females in low SDI areas, targeted genetic screening in high-prevalence zones, and primary care hearing service integration.

While our study benefits from the robust methodology of the GBD research, it still has limitations. The limited availability of detailed data prevents us from conducting a more detailed classification of ARoHL and performing a comprehensive risk factor analysis. Given the focus of our research, more detailed burden analyses at the national and subnational levels are necessary.

## Conclusions

The global ASPR of ARoHL increased by 5.63% (EAPC 0.163) and that of ASYR increased by 5.31% (EAPC 0.171) from 1990–2021, with the number of cases and YLDs doubling, confirming significant epidemiological progression. High-middle SDI regions presented the steepest increase in ASPR (EAPC 0.279), whereas East Asia accounted for 30% of the number of cases globally, with 6.07% ASPR growth. The recommended interventions include occupational noise control in rapid-burden regions, sex-specific rehabilitation for older females in low SDI areas, targeted genetic screening in high-prevalence zones, and primary care hearing service integration.

## Supporting information

**S1 File.   S1 Fig.** Global and regional trends in the ASPR and ASYR for ARoHL, 1990–2021. This figure illustrates the temporal trends in the ASPR and ASYR for ARoHL globally and across five SDI regions from 1990–2021, stratified by sex. These trends highlight the differential impact of ARoHL on various populations and underscore the importance of sex-specific analyses in understanding disease burden. (A) ASPR in both sexes; (B) ASPR in females; (C) ASPR in males; (D) ASYR in both sexes; (E) ASYR in females; (F) ASYR in males. **S2 Fig.** Correlation between the SDI and the ASPR/ASYR for ARoHL across 204 countries and territories in 2021. This figure illustrates the correlation between the SDI and the ASPR/ASYR for ARoHL across 204 countries and territories in 2021. This figure highlights the relationship between the SDI and the ARoHL burden, with scatter plots showing trends and correlations. Different colored dots represent different countries or territories. The gray curve represents the relationship between the SDI and the metric, with the shaded area indicating the 95% confidence interval. (A) SDI vs. ASPR; (B) SDI vs. ASYR. **S3 Fig.** Relationships between the EAPC and the ASPR/ASYR/SDI across 204 countries and territories, 1990–2021. This figure illustrates the relationships between the EAPCs (1990–2021) and the ASPR, ASYR, and SDI across 204 countries and territories. It includes scatter plots with regression lines and corresponding R and P values, indicating the temporal trends and sociodemographic influences on the ARoHL burden. Each dot represents a country or territory, with the size of the dot indicating the magnitude of the value. The blue curve shows the relationship between the X-axis and Y-axis parameters, with the shaded area representing the 95% confidence interval. (A) ASPR vs. its EAPC; (B) SDI vs. EAPC for ASPR; (C) ASYR vs. its EAPC; (D) SDI vs. EAPC for ASYR. **S4 Fig.** Frontier analysis. This figure identifies countries with the highest potential for improvement in the ARoHL burden on the basis of the ASPR and ASYR. This figure highlights regions where targeted public health interventions could effectively reduce the burden of ARoHL and improve health outcomes. Each dot in the figure represents a country. The frontier is depicted using a solid black line. In Figures A and C, the color gradient from light blue to dark blue represents the period from 1990–2021. In Figures B and D, red indicates a decrease, whereas blue indicates an increase. (A) SDI vs. ASPR by year; (B) SDI vs. ASPR by trends; (C) SDI vs. ASYR by year; (D) SDI vs. ASYR by trends. **S1 Table.** Age-standardized prevalence rate (ASPR) per 100,000 for age-related and other hearing loss in 1990 and 2021 and its estimated annual percentage change (EAPC) from 1990–2021 at the global and regional levels. **S2 Table**. Age-standardized YLD rate (ASYR) per 100,000 for age-related and other hearing loss in 1990 and 2021 and its estimated annual percentage change (EAPC) from 1990–2021 at the global and regional levels. **S3 Table**. Age-standardized prevalence rate (ASPR) per 100,000 for age-related and other hearing loss in 1990 and 2021 and its estimated annual percentage change (EAPC) from 1990–2021 across 204 countries and territories. **S4 Table**. Age-standardized YLD rate (ASYR) per 100,000 for age-related and other hearing loss in 1990 and 2021 and its estimated annual percentage change (EAPC) from 1990–2021 across 204 countries and territories. **S5 Table.**

Population-level determinants of changes in aging, population growth, and epidemiological changes in the number of cases and YLDs globally and by SDI quintile from 1990–2021.
(ZIP)

## Acknowledgments

The authors extend their sincere gratitude to all contributors of the Global Burden of Disease (GBD) Study 2021. They also express appreciation to the Institute for Health Metrics and Evaluation (IHME) at the University of Washington for providing unrestricted access to the GBD Study 2021 data.

## Author contributions

**Conceptualization:** Jiao Zhu, Deping Wang.

**Formal analysis:** Deping Wang.

**Methodology:** Jiao Zhu, Min Yang, Cuiying Zhou, Houyong Kang, Deping Wang.

**Supervision:** Deping Wang.

**Visualization:** Jiao Zhu, Deping Wang.

**Writing – original draft:** Jiao Zhu, Deping Wang.

**Writing – review & editing:** Jiao Zhu, Min Yang, Cuiying Zhou, Houyong Kang, Deping Wang.

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
