## [Decision Letter · Decision Letter 0]

16 Jun 2025

Dear Dr. Wang,

Thank you for submitting your manuscript to PLOS ONE. After careful consideration, we feel that it has merit but does not fully meet PLOS ONE’s publication criteria as it currently stands. Therefore, we invite you to submit a revised version of the manuscript that addresses the points raised during the review process.

**ACADEMIC EDITOR:  Please see the comments of the reviewers and revise the manuscript. **  

We look forward to receiving your revised manuscript.

Kind regards,

Gauri Mankekar, MD,PhD,FACS

Academic Editor

PLOS ONE

2. In the online submission form, you indicated that [The data for this study are available through the Global Health Data Exchange query tool (https://vizhub.healthdata.org/gbd-results/). All data reported in this article will be shared by the lead contact upon request.].

4. We note that Figure 3 in your submission contain [map/satellite] images which may be copyrighted. All PLOS content is published under the Creative Commons Attribution License (CC BY 4.0), which means that the manuscript, images, and Supporting Information files will be freely available online, and any third party is permitted to access, download, copy, distribute, and use these materials in any way, even commercially, with proper attribution. For these reasons, we cannot publish previously copyrighted maps or satellite images created using proprietary data, such as Google software (Google Maps, Street View, and Earth). For more information, see our copyright guidelines: http://journals.plos.org/plosone/s/licenses-and-copyright.

a. You may seek permission from the original copyright holder of Figure 3 to publish the content specifically under the CC BY 4.0 license. 

Reviewers' comments:

Reviewer's Responses to Questions

**Comments to the Author**

1. Is the manuscript technically sound, and do the data support the conclusions?

Reviewer #1: Yes

Reviewer #2: Partly

2. Has the statistical analysis been performed appropriately and rigorously?

Reviewer #1: Yes

Reviewer #2: Yes

3. Have the authors made all data underlying the findings in their manuscript fully available?

Reviewer #1: Yes

Reviewer #2: Yes

4. Is the manuscript presented in an intelligible fashion and written in standard English?

Reviewer #1: Yes

Reviewer #2: No

Reviewer #1: Observations

The study was done through a systematic literature analysis based on Global Burden of Disease (GBD) 2021 database. The study aimed at providing a comprehensive analysis of the global, regional, and national burden of Age Related and other Hearing Loss (ARoHL). The study involved the GBD 2021 database which integrates multidimensional epidemiological data from 204 countries and regions worldwide, covering 371 diseases and health issues, as well as 88 risk factors, spanning the years 1990 to 2021.

Overall, it is a well-thought out and executed research manuscript capable of contributing to the body of knowledge regarding the impact of ARoHL. However, I have the following observations,

1. Methods: The study utilized a variety of instruments such as the health inequality analysis. Based on the populations of countries like China and India, their contribution which had the largest impact on Slope index of Inequality, might have been due this factor. Hence comparism with smaller countries may not truly give realistic values. Hence might tend to have inherent sampling bias.

2. References: The referencing style appears generally fine. However, it appears the reference cited (references 9) due to its publication date, 1978 might not be reflective of current trends due to being slightly outdated.

3. The phrasing of the language used in the entire document could benefit from a more professional grammatical input.

4. I found the title, study design, statistics, figures and tables quite satisfactory.

Reviewer #2: Comments on Strengths.

a) Good graphs/figures with appropriate labels and annotations. Best figures are 1, 2, 4.

b) Compréhensive, quantitative analyses. Good use of techniques for decomposition analysis, cross-country inequality analysis, frontier analysis, and predictive analysis.

c) Attempt to provide a quantitative perspective for global, regional, and national prevalence of ARoHL that would be helpful for developing health-care plans appropriate for a given region/nation.

Issues, Items, and Questions to be Addressed.

General:

1) Provide a table with definitions for all acronyms used throughout the manuscript.

2) When giving numbers, show only the significant digits and use exponential notation when spanning several orders of magnitude. Do this throughout the text and also for Table 1 (which is difficult to read as it is).

3) Throughout the text, as a “standard”, state and use and discuss all variables in terms of normalized (for population size; i.e., per 100,000). When stating/discussion “raw”, un-normalized values, then clearly state that (and distinguish from normalized values).

Specific:

1) The reference list is difficult to read. Re-format. For example, for a given reference indent all lines after the first. In addition, double space between references.

2) Table 1 is difficult to read. Re-format per above. Consider putting this into supplementary material (and relying on the figures).

3) Abstract

a. Emphasize prevalence and YLDs that are normalized for population size (e.g., cases per fixed number of people within a given population).

b. Emphasize trends, disparities, and differences between using normalized values (for population size) for prevalence an YLDs.

4) Add more detail, emphasis, and discussion about normalizing the prevalence, YLDs, and distribution of ARoHL with respect to the population size at each time point being analyzed. In other words, prevalence per fixed number of people within a given population. The normalized variables are more meaningful in terms of explaining possible effects of the various “driving” factors (compared to the effect of increases/decreases in the total population size on the raw, absolute numbers).

5) Provide some examples of possible actions that could be taken and strategies that could be proposed, in view of the projections for future trends.

6) Decomposition Analysis

a. Discuss possible other/confounding factors, including the following: noise-induced hearing loss (outside environment and home/work conditions), side-effects of medical treatments (medications/drugs, surgery, radiation), family history and genetic pre-dispositions.

b. Line 134: Provide a table listing all the factors “behind changes in the disease burden of ARoHL” and their percent contributions. Emphasize the percent contributions for changes in a normalized (for population size) prevalence of ARoHL.

c. Line 140: Discuss “the accelerated aging process”.

d. Line 147: Explain and discuss the meaning of “for a comprehensive understanding of the epidemiological evolution of middle ear disease”, within the context of ARoHL. Why refer to “middle ear disease”?

7) Results

a. Emphasize ASPR per 100,000 people within a given population (i.e., normalized for population size).

b. Lines 216 - 218: State that these numbers for prevalence are a raw number of cases in the population. State that the ASPR is per 100,000 (i.e., normalized for population size. It is especially meaningful to emphasize changes in a normalized prevalence (cases per fixed number of people within a population) over the 21-year time period. Thereafter, one can consider possible factors involved (other than population increase or decrease).

c. Lines 218 – 219: In regard to “age-standardized prevalence rate (ASPR)”, provide a literature reference/citation that details/describes comparing across populations with different age structures (using weights based on the age distribution of a standard population).

d. Table 1 legend: State the EAPC is based on the normalized ASPRs (per 100,000 people).

e. Figure 1 legend: State normalized ASPR and normalized YLDs are shown in the figure.

8) Conclusions

a. Lines 448 – 452: Expand. Besides “raw”, un-normalized total numbers, emphasize changes in numbers that are normalized for population size. Summarize the percent changes (for normalized data). This way one can meaningfully consider changes in prevalence related to all factors (other than population growth) which might be important for considering intervention strategies. Finally, suggest (speculate) possible intervention strategies.

**Do you want your identity to be public for this peer review?** For information about this choice, including consent withdrawal, please see our Privacy Policy

Reviewer #1: **Yes: ** Stephen O Adebola

Reviewer #2: **Yes: ** John H Anderson, MD, PhD

---

## [Author Response · Author response to Decision Letter 1]

3 Jul 2025

We have uploaded our detailed responses to the reviewers' comments as a separate document titled 'Response to Reviewers.' The specific contents are as follows:

Reviewer #1:

1. Methods: The study utilized a variety of instruments such as the health inequality analysis. Based on the populations of countries like China and India, their contribution which had the largest impact on Slope index of Inequality, might have been due this factor. Hence comparism with smaller countries may not truly give realistic values. Hence might tend to have inherent sampling bias.

Response:

We acknowledge the limitation regarding the influence of large-population countries on the Slope Index of Inequality. Unfortunately, there is no universally accepted method to fully address this issue

2. References: The referencing style appears generally fine. However, it appears the reference cited (references 9) due to its publication date, 1978 might not be reflective of current trends due to being slightly outdated.

Response:

We have replaced the outdated reference (Reference 10) with a more recent and relevant source to better reflect current trends.

3. The phrasing of the language used in the entire document could benefit from a more professional grammatical input.

Response:

Taking your suggestions into consideration, we have had the language of the entire document polished by AJE.

4. I found the title, study design, statistics, figures and tables quite satisfactory.

Response:

We appreciate the reviewer's positive feedback on the title, study design, statistics, figures, and tables. However, after revising and polishing the entire manuscript, we have made minor adjustments to these sections to ensure overall consistency.

Reviewer #2:

General:

1) Provide a table with definitions for all acronyms used throughout the manuscript.

Response:

In response to your suggestion, a “List of Abbreviations” has been added immediately following the Conclusion section. This table provides definitions for all acronyms and abbreviations used throughout the manuscript to enhance clarity.

2) When giving numbers, show only the significant digits and use exponential notation when spanning several orders of magnitude. Do this throughout the text and also for Table 1 (which is difficult to read as it is).

Response:

We have revised the numerical data presentation throughout the text to show only significant digits and used exponential notation where appropriate. Additionally, Table 1 has been revised and moved to the Supporting Information as S1 Table. We have also applied the same formatting changes to S2 Table.

3) Throughout the text, as a “standard”, state and use and discuss all variables in terms of normalized (for population size; i.e., per 100,000). When stating/discussion “raw”, un-normalized values, then clearly state that (and distinguish from normalized values).

Response:

We have standardized the presentation of all variables throughout the manuscript using normalized values (per 100,000) according to this suggestion. For ASPR and ASYR, we have explicitly noted that these are expressed per 100,000. Additionally, we have clearly distinguished raw, un-normalized values by specifying them as “number of cases” or “number of YLDs” whenever they appear. This ensures consistency and clarity in our data presentation.

Specific:

1) The reference list is difficult to read. Re-format. For example, for a given reference indent all lines after the first. In addition, double space between references.

Response:

We have reformatted the references in accordance with your request.

2) Table 1 is difficult to read. Re-format per above. Consider putting this into supplementary material (and relying on the figures).

Response:

We have re-formatted Table 1 as requested and renamed it as S1 Table, placing it in the Supporting Information.

3) Abstract

a. Emphasize prevalence and YLDs that are normalized for population size (e.g., cases per fixed number of people within a given population).

Response:

In the abstract, we have explicitly noted that ASPR and ASYR are expressed per 100,000, emphasizing the use of normalized values as requested.

b. Emphasize trends, disparities, and differences between using normalized values (for population size) for prevalence an YLDs.

Response:

We have revised the abstract to emphasize the trends, disparities, and differences in prevalence and YLDs when using normalized values (per 100,000).

4) Add more detail, emphasis, and discussion about normalizing the prevalence, YLDs, and distribution of ARoHL with respect to the population size at each time point being analyzed. In other words, prevalence per fixed number of people within a given population. The normalized variables are more meaningful in terms of explaining possible effects of the various “driving” factors (compared to the effect of increases/decreases in the total population size on the raw, absolute numbers).

Response:

We have revised the manuscript to emphasize and discuss the normalization of prevalence, YLDs, and ARoHL distribution with respect to population size at each time point, focusing primarily on the normalized variables as requested.

5) Provide some examples of possible actions that could be taken and strategies that could be proposed, in view of the projections for future trends.

Response:

We have incorporated examples of possible actions and strategies that could be proposed based on the projections for future trends, as requested.

6) Decomposition Analysis

a. Discuss possible other/confounding factors, including the following: noise-induced hearing loss (outside environment and home/work conditions), side-effects of medical treatments (medications/drugs, surgery, radiation), family history and genetic pre-dispositions.

Response:

We have added a discussion of additional potential confounding factors, including noise-induced hearing loss, medical treatment side effects, and genetic predispositions, in the discussion section.

b. Line 134: Provide a table listing all the factors “behind changes in the disease burden of ARoHL” and their percent contributions. Emphasize the percent contributions for changes in a normalized (for population size) prevalence of ARoHL.

Response:

We have detailed the drivers of absolute changes in ARoHL burden in Supplementary Table S5, quantifying contributions from population growth, aging, and epidemiological shifts. For normalized prevalence (ASPR), given its design to remove population size effects, we currently believe that attributing specific drivers would be methodologically unsound. Instead, we have analyzed the relationships between ASPR and factors such as age, gender, and SDI throughout the text.

c. Line 140: Discuss “the accelerated aging process”.

Response:

We apologize for the oversight in our manuscript where we inadvertently used the term “the accelerated aging process” instead of “population aging.” We have now corrected this error. While we acknowledge the importance of discussing population aging, we did not elaborate further on this topic in our discussion, as population aging is difficult to change in the short term, as mentioned in our discussion section.

d. Line 147: Explain and discuss the meaning of “for a comprehensive understanding of the epidemiological evolution of middle ear disease”, within the context of ARoHL. Why refer to “middle ear disease”?

Response:

We appreciate the reviewer’s attention to this point. The reference to “middle ear disease” was indeed an oversight, and we intended to discuss ARoHL. We were concurrently working on a study related to middle ear disease, which likely contributed to the confusion. We have now corrected this error and have thoroughly reviewed the entire manuscript to ensure consistency and accuracy, minimizing the possibility of similar mistakes.

7) Results

a. Emphasize ASPR per 100,000 people within a given population (i.e., normalized for population size).

Response:

We have made the revision as requested.

b. Lines 216 - 218: State that these numbers for prevalence are a raw number of cases in the population. State that the ASPR is per 100,000 (i.e., normalized for population size. It is especially meaningful to emphasize changes in a normalized prevalence (cases per fixed number of people within a population) over the 21-year time period. Thereafter, one can consider possible factors involved (other than population increase or decrease).

Response:

We have made the revision as requested.

c. Lines 218 – 219: In regard to “age-standardized prevalence rate (ASPR)”, provide a literature reference/citation that details/describes comparing across populations with different age structures (using weights based on the age distribution of a standard population).

Response:

We have addressed this point by providing a literature reference and detailed explanation regarding the calculation of the age-standardized prevalence rate (ASPR) and age-standardized YLDs rate (ASYR) in the “Estimation of disease burden” section of the Materials & Methods.

d. Table 1 legend: State the EAPC is based on the normalized ASPRs (per 100,000 people).

Response:

We have made the revision as requested.

e. Figure 1 legend: State normalized ASPR and normalized YLDs are shown in the figure.

Response:

We have made the revision as requested.

8) Conclusions

a. Lines 448 – 452: Expand. Besides “raw”, un-normalized total numbers, emphasize changes in numbers that are normalized for population size. Summarize the percent changes (for normalized data). This way one can meaningfully consider changes in prevalence related to all factors (other than population growth) which might be important for considering intervention strategies. Finally, suggest (speculate) possible intervention strategies.

Response:

We have made the revision as requested.

---

## [Decision Letter · Decision Letter 1]

5 Aug 2025

Global burden and trends of age-related and other hearing loss: A 32-year analysis and future projections based on the GBD 2021

PONE-D-25-18205R1

Dear Dr. Wang,

We’re pleased to inform you that your manuscript has been judged scientifically suitable for publication and will be formally accepted for publication once it meets all outstanding technical requirements.

Kind regards,

Gauri Mankekar, MD,PhD,FACS

Academic Editor

PLOS ONE

Additional Editor Comments (optional):

Reviewers' comments:

Reviewer's Responses to Questions

**Comments to the Author**

Reviewer #1: (No Response)

Reviewer #2: All comments have been addressed

2. Is the manuscript technically sound, and do the data support the conclusions?

Reviewer #1: Yes

Reviewer #2: Yes

3. Has the statistical analysis been performed appropriately and rigorously?

Reviewer #1: Yes

Reviewer #2: Yes

4. Have the authors made all data underlying the findings in their manuscript fully available?

Reviewer #1: Yes

Reviewer #2: Yes

5. Is the manuscript presented in an intelligible fashion and written in standard English?

Reviewer #1: Yes

Reviewer #2: Yes

Reviewer #1: Observations

The study was done through a systematic literature analysis based on the Global Burden of Disease (GBD) 2021 database. The study aimed at providing a comprehensive analysis of the global, regional, and national burden of Age Related and other Hearing Loss (ARoHL). The study involved the GBD 2021 database which integrates multidimensional epidemiological data from 204 countries and regions worldwide, covering 371 diseases and health issues, as well as 88 risk factors, spanning the years 1990 to 2021.

Overall, it is a well-thought out and executed research manuscript capable of contributing to the body of knowledge regarding the impact of ARoHL. However, I have the following observations,

1. Methods: Study design appeared a bit skewed, but acceptable in current version

2. References: Improved upon based on updated revised copy

3. The phrasing of the language reads more grammatically based on professional input.

I found the title, study design, statistics, figures and tables quite satisfactory

Reviewer #2: The reviewers' requests were fully addressed. The clarity, specificity, and context for the results have been significantly improved.

**Do you want your identity to be public for this peer review?** For information about this choice, including consent withdrawal, please see our Privacy Policy

Reviewer #1: **Yes: ** Stephen Adebola

Reviewer #2: **Yes: ** John H Anderson

---

## [Editor Report · Acceptance letter]

PONE-D-25-18205R1

PLOS ONE

Dear Dr. Wang,

I'm pleased to inform you that your manuscript has been deemed suitable for publication in PLOS ONE. Congratulations! Your manuscript is now being handed over to our production team.

Kind regards,

on behalf of

Dr. Gauri Mankekar

Academic Editor

PLOS ONE